# Association between the Body Roundness Index and osteoarthritis: A population-based study

**Yue Qiu[1◉], Huangyi Yin[2◉], Jinzhi Meng[1], Yang Cai[1], Jun Yao [1]\***

**1** Osteoarticular Surgery, The First Affiliated Hospital of Guangxi Medical University, Nanning, China,
**2** Geriatric Endocrinology, The First Affiliated Hospital of Guangxi Medical University, Nanning, China

◉ These authors contributed equally to this work.
\* yaojun800524@126.com

## Abstract

### Background

Obesity is recognized as an independent risk factor for the development of osteoarthritis (OA). The Body Roundness Index (BRI) represents a novel metric for assessing obesity, yet its connection to OA remains unclear. This research aims to explore the potential association between BRI and OA.

### Methods

We analyzed data from 20,564 participants who were part of the National Health and Nutrition Examination Survey (NHANES) conducted between 2007 and 2016. The association between BRI and the prevalence of OA was examined using multifactorial logistic regression and restricted cubic spline (RCS) analysis. To ensure the reliability of the findings, we conducted a stratified analysis.

### Results

The average BRI among the participants was 5.18 (0.03), with an OA prevalence of 11.98%. Following adjustment for all covariates, continuous BRI exhibited a significant positive association with OA (OR: 1.19, 95% CI: 1.15–1.24, $P < 0.0001$). Participants in the top quartile of BRI showed a 153% higher prevalence of OA relative to individuals in the lowest quartile (OR: 2.53, 95% CI: 2.01–3.19, $P < 0.0001$). The RCS curve demonstrated a linear relationship between BRI and OA. Subgroup analysis examined that the observed relationship was significant exclusively among individuals with a BMI ≥ 30 kg/m². This association remained unaffected by variables such as race, age, gender, hypertension, cardiovascular disease (CVD), or diabetes.

**Data availability statement:** All the data are publicly accessible in Nhanes database (https://www.cdc.gov/nchs/nhanes/index.htm). Others can access this data in the same way that authors use the information in our Data Availability statement. And the author does not

have special access to the data that anyone else does not have.

**Funding:** This work was supported by grants from the Health Department of Guangxi Zhuang Autonomous Region Self-funded project (grant/award number: Z2013039). And Jun Yao was involved in obtaining funding, supervising, writing-reviewing, and editing in this study.

**Competing interests:** The authors have declared that no competing interests exist.

**Abbreviations:** OA, osteoarthritis; BMI, Body Mass Index; WC, waist circumference; METS-VF, Metabolic Score for Visceral Fat; VAT, visceral adipose tissue; BRI, Body Roundness Index; NHANES, National Health and Nutrition Examination Survey; NCHS, National Center for Health Statistics; PIR, poverty-to-income ratio; PA, physical activity; MEC, Mobile Examination Center; MET, metabolic equivalents; TC, total cholesterol; HDL-C, high-density lipoprotein cholesterol; ALT, alanine aminotransferase; AST, aspartate aminotransferase; CVD, cardiovascular disease; RCS, restricted cubic spline; ROC, receiver operating characteristic; KOA, knee osteoarthritis; ECM, extracellular matrix; TNF-α, tumor necrosis factor-alpha; IL-6, interleukin-6; MMP-13, matrix metalloproteinase 13; ADAMTS-4, activity of a disintegrin and metalloproteinase with thrombospondin motifs 4; FLS, fibroblast-like synoviocytes; OR, odds ratio.

## Conclusion

An increased BRI is associated with a higher prevalence of OA, particularly in obese populations. BRI is expected to become a valuable indicator for identifying individuals at high risk of OA.

## Introduction

Osteoarthritis (OA) is a prevalent chronic degenerative condition that primarily affects the articular cartilage and surrounding soft tissues of the joints [1]. As of 2019, approximately 500 million individuals had been diagnosed with OA, and this figure is anticipated to grow. The global incidence of OA is forecasted to reach 1 billion by 2035 [2]. In the United States, annual healthcare costs related to OA exceed $10 billion, imposing a notable socioeconomic burden [3]. OA significantly contributes to disability and diminished quality of life among older adults, with its progression linked to factors including age, gender, obesity, genetic factors, diet [4–7]. Identifying and managing early risk factors for OA is crucial to mitigating its risk, underscoring the importance of reliable methods to pinpoint individuals at elevated risk of developing the condition.

Obesity, which affects over 1 billion people worldwide, is considered a global epidemic with expected increases in prevalence. By 2030, more than 14% of men and over 20% of women globally are projected to be obese [8]. This trend is notably pronounced in older populations. A decade-long health survey in Scotland found that while overall obesity rates have remained relatively stable, there has been a noticeable rise in BMI among individuals aged 60–70 [9]. Furthermore, in Europe, obesity rates among individuals aged 60 and above are estimated to range from 20–30%, while in the United States, the figure exceeds 35% [10]. Given that OA is one of the common age-related diseases, the relationship between OA and obesity has received considerable research focus. Evidence indicates that obesity not only affects the likelihood of OA but also significantly impacts the severity of the disease [10–13]. Mendelian randomization studies have suggested that increased Body Mass Index (BMI) is a notable factor in the heightened risk of hip OA [14]. The pathways through which obesity elevates OA risk are intricate, involving not just increased mechanical stress but also the secretion of various adipokines and inflammatory mediators [15–17].

Obesity is undoubtedly critical in the development of OA. Traditionally, obesity assessments have relied on indicators such as BMI and waist circumference (WC). However, recent studies suggest that BMI may not fully represent metabolic health or fat distribution, potentially leading to an inaccurate assessment of the adverse effects of fat accumulation on overall health [18,19]. While WC is a standard measure for abdominal obesity, it overlooks height and does not differentiate between visceral and subcutaneous fat [20,21]. To address these limitations, Thomas et al. introduced the Body Roundness Index (BRI), which calculates body roundness using an ellipsoidal model and estimates visceral and total body fat percentages through eccentricity [22]. As a novel obesity metric, BRI offers a more thorough view of visceral fat compared to traditional measurements. BRI has proven to be more comprehensive than conventional metrics in

diagnosing specific conditions, such as gallstones, overactive bladder, kidney stones, and depression [20,23–25]. Furthermore, research found that the Metabolic Score for Visceral Fat (METS-VF) is a more reliable predictor of OA than BMI [26]. Mendelian randomization research indicated a 40% increase in OA risk for each unit rise in visceral adipose tissue (VAT) [18]. VAT promotes fat synthesis and breakdown, leading to increased production of pro-inflammatory cytokines, which further damage articular cartilage [27,28]. This suggests that visceral fat accumulation may significantly influence the pathological process of OA. Despite the BRI focus on visceral fat, its potential link to OA is not well established. Thus, we propose that the BRI might be positively related to OA prevalence and offer enhanced diagnostic capabilities compared to traditional body measures.

This study intends to assess the relationship between BRI and OA by utilizing the National Health and Nutrition Examination Survey (NHANES) and to assess its efficacy in comparison to traditional body metrics, seeking a simpler and more efficient method for OA risk identification.

## Methods

### Data sources

NHANES is a comprehensive longitudinal study managed by the National Center for Health Statistics (NCHS). This study assessed participants' physical condition using questionnaires, interviews, laboratory tests, and physical examinations. Informed consent was received from all individuals involved. This survey was approved by the NCHS Ethics Review Board, and all methods of this study were conducted in accordance with relevant guidelines and regulations.

### Study population and design

Given the constraints of the NHANES questionnaire, we utilized data spanning from 2007 to 2016, encompassing a substantial sample of 50,588 individuals. After applying specific inclusion criteria, we ultimately selected 20,564 participants through the following process: 1) exclusion of participants under 20 years of age (N = 21,387) or those who were pregnant (N = 318); 2) removal of participants lacking data on height, weight, and WC (N = 2,770); 3) exclusion of individuals who either did not complete the OA Questionnaire (N = 2,223) or were diagnosed with other types of arthritis (N = 2,110); and 4) omission of participants with incomplete covariate data (N = 1,216). Participants with incomplete data regarding physical activity (PA), alcohol consumption, and poverty-to-income ratio (PIR) were included in the "unknown" group (Fig 1). Subsequently, the eligible participants were divided into OA and non-OA groups for analysis, with the aim of thoroughly investigating the potential relationship between BRI and OA.

### Assessment of body measurement indicators

WC, height, and weight measurements were obtained at the Mobile Examination Center (MEC), with subjects instructed to wear light clothing and remove their footwear. The BMI and BRI were computed using established formulas:

$$BMI = \frac{weight\ (kg)}{height\ (cm)^2}$$

$$BRI = 364.2 - 365.5 \times \sqrt{1 - \frac{\left(\frac{WC}{2\pi}\right)^2}{(0.5\ height\ )^2}}$$

### Diagnosis of osteoarthritis

Previous research indicates that the diagnosis of OA in NHANES participants can be confirmed through questionnaire responses [29,30]. Specifically, participants who self-reported being diagnosed with arthritis by a healthcare professional, and identified the type of arthritis as OA, were considered to have OA.

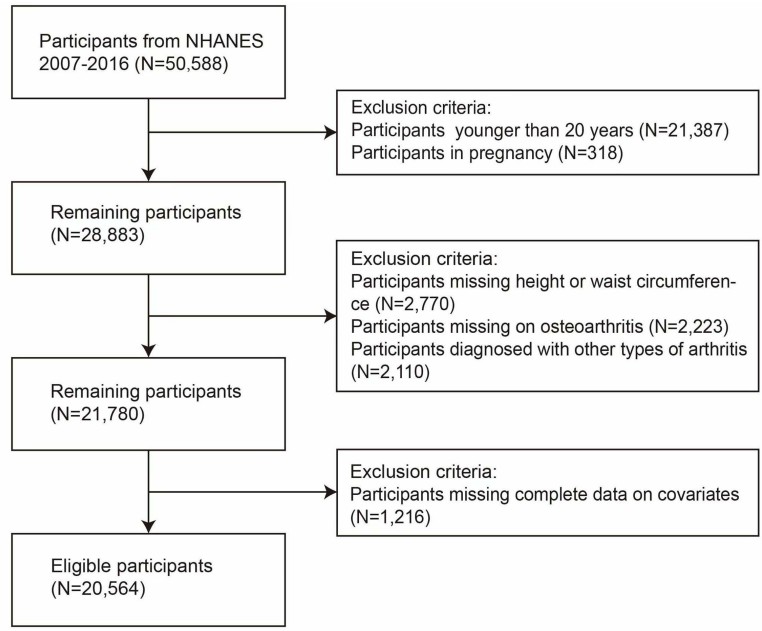

**Fig 1. Screening process for qualified participants.**

## Covariables

Building on existing studies, this research identified several potential covariables to minimize bias in the results. Data on race, age, sex, marital status, smoking, alcohol consumption, PA, education, and PIR were collected through interviews. PA was quantified in metabolic equivalents (MET) and categorized into high activity (≥600 MET-min/week), low activity (<600 MET-min/week), and unknown groups. In addition, NHANES staff collect blood samples at the MEC to measure high-density lipoprotein cholesterol (HDL-C), uric acid, total cholesterol (TC), total calcium, blood creatinine, alanine aminotransferase (ALT), and aspartate aminotransferase (AST) levels. Participants will be considered to meet the diabetes diagnosis criteria if they meet any of the following conditions: HbA1c ≥ 6.5%, fasting blood glucose ≥ 7.0 mmol/l, 2-hour postprandial blood glucose ≥ 11.1 mmol/l, currently taking hypoglycemic medications or using insulin, or self-reporting a history of diabetes. Cardiovascular disease (CVD) diagnosis was determined from self-reported history of conditions. Hypertension was diagnosed if participants were either on blood pressure-lowering medication, had been diagnosed with hypertension by a healthcare provider, or exhibited high blood pressure during onsite assessments.

## Statistical analysis

This study incorporated the official NHANES-recommended weights, allowing for the generalization of the results to the U.S. non-institutionalized population. A t-test evaluated differences in continuous variables between groups, with results shown as means and standard errors. For categorical variables, the chi-square test was applied, presenting results as frequencies and percentages. The possible relationship between BRI and OA was investigated using three multifactor logistic regression models. Model 2 adjust for race, gender, and age, while Model 1 did not adjust for any covariates. Model 3 included all potential variables such as race, age, sex, marital status, PA, education, smoking, alcohol consumption, PIR, blood uric acid, total calcium, blood creatinine, TC, HDL-C, ALT, AST, diabetes, CVD, and hypertension. The restricted cubic spline (RCS) curve visualized the relationship between BRI and OA, while the receiver operating characteristic

(ROC) curve compared the role of BRI with other traditional obesity indicators in the diagnosis of OA. Subgroup analyses across gender, age, race, BMI, hypertension, CVD, and diabetes evaluated the robustness of this association.

## Results

### Comparison of baseline characteristics of participants

Follow the exclusion criteria, 20,564 participants were included, with 50.05% male and an average age of 45.61 (0.26). Among them, 2,464 met OA diagnostic criteria, with an average BRI of 5.18 (0.03). Significant differences in gender, age, marital status, education level, alcohol consumption, PIR, smoking, PA, HDL-C, TC, ALT, and creatinine were noted between OA and non-OA groups. OA patients showed higher tendencies for CVD, hypertension, diabetes, and had greater WC, weight, BMI, and BRI compared to non-OA individuals. Table 1 details these baseline characteristics.

### Association between BRI and the prevalence of osteoarthritis

As detailed in Table 2, BRI was positively related with OA in Model 3 (OR: 1.19, 95% CI: 1.15–1.24, $P < 0.0001$). After accounting for all potential influencing factors, participants in the highest BRI quartile had a 1.48-fold increased likelihood of OA compared to individuals in the lowest quartile (OR: 2.53, 95% CI: 2.01–3.19, $P < 0.0001$). A positive association between BRI and the prevalence of OA was also identified in the RCS analysis ($P$ overall $< 0.0001$), with no evidence of nonlinearity ($P$ nonlinear $= 0.258$), as illustrated in Fig 2.

### Comparison of the accuracy of the BRI and other body measures in diagnosing OA

The ROC curves were used to compare the diagnostic performance of BRI and other traditional body measurements (WC, weight, BMI) for OA. As shown in Fig 3, the AUC values for BRI, WC, BMI, and weight were 0.653, 0.624, 0.595, and 0.545, respectively. These results indicate that BRI showed a marginally higher AUC value compared to the other measures, although the overall performance of all parameters was modest.

### Subgroup analysis

We conducted subgroup analyses based on factors such as race, age, gender, BMI, CVD, hypertension and diabetes. As shown in Fig 4, the positive relationship between BRI and OA remained unaffected by sex, age, race, hypertension, CVD, and diabetes ($P$ for interaction >0.05). A notable interaction was observed between BRI and BMI, revealing a considerable positive relationship exclusively among individuals classified as obese (BMI ≥ 30 kg/m²). Specifically, an increment of one unit in BRI corresponded to a 13% higher prevalence of OA among individuals with a BMI of 30 kg/m² or more (OR: 1.13, 95% CI: 1.07–1.20, $P < 0.001$).

## Discussion

This research represents the first investigation into the potential link between BRI and OA. In this cross-sectional analysis, a higher BRI was related with a greater prevalence of OA, particularly in individuals with a BMI of 30 kg/m² or greater. RCS curves showed that BRI levels were linearly and positively associated with the prevalence of OA. In addition, ROC analysis suggested that BRI showed a marginally higher AUC value compared to traditional indicators (BMI, WC, and weight) for assessing OA.

OA is widely prevalent in the elderly population and is a major contributor to disability among the elderly. With the accelerated aging of the population, OA presents a range of public health and economic challenges that are becoming increasingly prominent. Obesity is a global epidemic, particularly affecting the elderly. Therefore, the association between obesity and OA, as a common aging-related disease, is compelling. Numerous studies have documented the association between obesity and OA [31–33]. A study reported that individuals with a BMI of more than 30 kg/m2 had a 6.8-fold higher risk of

**Table 1. Weighted comparison of baseline characteristics.**

| Variables | Total (N = 20,564) | Non-OA (N = 18,100) | OA (N = 2,464) | P-value |
|---|---|---|---|---|
| OA (%) | | | | – |
| No | 18100(88.02) | 18100(88.02) | – | |
| Yes | 2464(11.98) | – | 2464(11.98) | |
| Age (years) | 45.61(0.26) | 43.22(0.25) | 61.50(0.30) | < 0.0001 |
| Sex (%) | | | | < 0.0001 |
| Female | 10188(49.95) | 8615(47.75) | 1573(64.52) | |
| Male | 10376(50.05) | 9485(52.25) | 891(35.48) | |
| Race (%) | | | | < 0.0001 |
| Non-Hispanic Black | 3931(10.16) | 3575(10.81) | 356(5.87) | |
| Non-Hispanic White | 8585(66.92) | 7038(64.36) | 1547(83.88) | |
| Mexican American | 3321(9.08) | 3111(9.99) | 210(3.06) | |
| Other | 4727(13.84) | 4376(14.85) | 351(7.19) | |
| Marital status (%) | | | | < 0.0001 |
| Never married | 4110(19.40) | 3928(21.31) | 182(6.72) | |
| Separated | 3952(16.35) | 3135(14.72) | 817(27.19) | |
| Married | 12502(64.25) | 11037(63.97) | 1465(66.08) | |
| Education level (%) | | | | 0.049 |
| Below high school | 2000(5.12) | 1786(5.26) | 214(4.21) | |
| High school graduate | 7387(31.76) | 6534(31.97) | 853(30.40) | |
| Above high school | 11177(63.12) | 9780(62.78) | 1397(65.39) | |
| PIR (%) | | | | < 0.0001 |
| < 1.3 | 5912(19.79) | 5315(20.52) | 597(14.94) | |
| 1.3-3.5 | 6953(32.47) | 6059(32.24) | 894(33.95) | |
| ≥ 3.5 | 5983(41.12) | 5191(40.54) | 792(44.95) | |
| Unknown | 1716(6.62) | 1535(6.69) | 181(6.15) | |
| PA (%) | | | | < 0.0001 |
| Low | 2761(13.19) | 2375(12.86) | 386(15.44) | |
| High | 12907(66.86) | 11671(68.62) | 1236(55.19) | |
| Unknown | 4896(19.95) | 4054(18.52) | 842(29.37) | |
| Smoking (%) | | | | < 0.0001 |
| Never | 11819(57.02) | 10655(58.51) | 1164(47.14) | |
| Former | 4534(23.19) | 3663(21.21) | 871(36.30) | |
| Now | 4211(19.78) | 3782(20.27) | 429(16.55) | |
| Drinking (%) | | | | < 0.0001 |
| Never | 2695(10.24) | 2360(10.24) | 335(10.22) | |
| Former | 3043(12.63) | 2467(11.50) | 576(20.13) | |
| Mild | 6186(33.19) | 5291(32.06) | 895(40.65) | |
| Moderate | 2950(16.31) | 2626(16.48) | 324(15.20) | |
| Heavy | 4132(20.97) | 3911(22.69) | 221(9.60) | |
| Unknown | 1558(6.66) | 1445(7.03) | 113(4.20) | |
| Diabetes (%) | | | | < 0.0001 |
| No | 17290(87.94) | 15479(89.42) | 1811(78.16) | |
| Yes | 3274(12.06) | 2621(10.58) | 653(21.84) | |
| Hypertension (%) | | | | < 0.0001 |
| No | 12970(66.61) | 12126(70.77) | 844(39.06) | |
| Yes | 7594(33.39) | 5974(29.23) | 1620(60.94) | |

*(Continued)*

**Table 1.** (Continued)

| Variables | Total (N = 20,564) | Non-OA (N = 18,100) | OA (N = 2,464) | *P*-value |
|---|---|---|---|---|
| CVD (%) | | | | < 0.0001 |
| No | 18913(93.39) | 16947(94.99) | 1966(82.79) | |
| Yes | 1651(6.61) | 1153(5.01) | 498(17.21) | |
| TC (mmol/l) | 5.00(0.01) | 4.99(0.01) | 5.12(0.03) | < 0.0001 |
| HDL-C (mmol/l) | 1.38(0.01) | 1.37(0.01) | 1.44(0.02) | < 0.0001 |
| ALT (U/L) | 25.75(0.16) | 26.00(0.18) | 24.08(0.30) | < 0.0001 |
| AST (U/L) | 25.79(0.13) | 25.78(0.13) | 25.83(0.38) | 0.907 |
| Uric acid (umol/l) | 321.59(0.86) | 321.27(0.89) | 323.72(2.27) | 0.301 |
| Total calcium (mmol/l) | 2.35(0.00) | 2.35(0.00) | 2.36(0.00) | 0.097 |
| Creatinine (umol/l) | 77.50(0.27) | 77.25(0.29) | 79.17(0.58) | 0.003 |
| WC (cm) | 98.07(0.24) | 97.17(0.24) | 103.97(0.45) | < 0.0001 |
| Weight (kg) | 81.84(0.25) | 81.40(0.25) | 84.78(0.58) | < 0.0001 |
| BMI (kg/m²) | 28.54(0.09) | 28.24(0.09) | 30.54(0.21) | < 0.0001 |
| BRI | 5.18(0.03) | 5.03(0.03) | 6.21(0.07) | < 0.0001 |

Categorical variables were analyzed using the chi-square test, while comparisons of continuous variables were conducted using the t-test.

OA: osteoarthritis; PIR: poverty income ratio; PA: physical activity; CVD: cardiovascular disease; TC: total cholesterol; HDL-C: high-density lipoprotein cholesterol; ALT: Alanine aminotransferase; AST: Aspartate aminotransferase; WC: waist circumference; BMI: body mass index.

knee OA (KOA) than non-obese subjects [12]. Meanwhile, a biomechanical study of obese patients showed that a higher BMI implies greater joint loading in the lower extremities, which can increase the risk of OA [17]. Additionally, evidence from in vivo studies suggests that compressive loading of the knee joint leads to cartilage fibrosis and erosion as well as osteophyte formation [34]. Under the combined influence of OA and obesity, which leads to decreased exercise ability, reduced activity, further weight gain, and decreased muscle strength, a vicious cycle ensues that accelerates the progression of OA [35]. Conversely, weight loss is significant in the prevention and treatment of OA. In a cohort study of women, a reduction in BMI by more than 2 units reduced the risk of symptomatic KOA by at least 50%, a finding also applicable to populations with a baseline BMI ≥ 25 kg/m² [6]. Another cohort study of 1,410 individuals with symptomatic KOA demonstrated a significant dose-response relationship between weight change and both joint pain and physical function scores, showing that reducing weight could help alleviate symptoms of KOA [36].

The mechanisms by which obesity promotes the advancement and development of OA are complex. Traditionally, it is believed that obesity leads to increased mechanical loading of the joints, which activates the inflammatory response, induces oxidative stress, triggers chondrocyte apoptosis, and causes extracellular matrix (ECM) degradation, ultimately leading to the development of OA [37–39]. However, the mechanical loading hypothesis alone does not sufficiently explain the relationship between obesity and non-weight-bearing joint OA [40]. This effect may partly result from the secretion of inflammatory factors and adipokines from adipose tissue. As one of the most metabolically active tissues, adipose tissue generates substantial quantities of inflammatory mediators, such as interleukin-6 (IL-6) and tumor necrosis factor-alpha (TNF-α), through its metabolic activities [41]. IL-6 promotes cartilage degradation in OA by upregulating matrix metalloproteinase 13 (MMP-13) and aggregated protein polymerase expression [42]. TNF-α promotes the catabolism of proteoglycans and type II collagen by activating the expression and activity of a disintegrin and metalloproteinase with thrombospondin motifs 4 (ADAMTS-4), further destroying the integrity of the cartilage extracellular matrix and promoting articular cartilage decomposition [43]. In vitro experiments have also shown that adipose tissue from the infrapatellar fat pad in end-stage KOA acts on fibroblast-like synoviocytes (FLS) to secrete inflammatory factors, inducing a synovial

**Table 2. Weighted logistic regression for association between BRI and OA.**

| Exposures | Model1 [OR (95% CI) *P*-value] | Model2 [OR (95% CI) *P*-value] | Model3 [OR (95% CI) *P*-value] |
|---|---|---|---|
| BRI (Continuous) | 1.23(1.20,1.26) <0.0001 | 1.19(1.15,1.22) <0.0001 | 1.19(1.15,1.24) <0.0001 |
| BRI (Quartiles) | | | |
| Q1 (≤3.73) | ref | ref | ref |
| Q2 (3.73–4.96) | 1.98(1.64,2.38) <0.0001 | 1.37(1.13,1.67) 0.002 | 1.37(1.11,1.70) 0.004 |
| Q3 (4.96–6.45) | 2.67(2.22,3.21) <0.0001 | 1.52(1.25,1.86) <0.0001 | 1.50(1.21,1.87) <0.001 |
| Q4 (>6.45) | 4.45(3.78,5.24) <0.0001 | 2.58(2.17,3.06) <0.0001 | 2.53(2.01,3.19) <0.0001 |
| *P* for trend | <0.0001 | <0.0001 | <0.0001 |

Model 1: Adjusted for no variables.

Model 2: Adjusted for race, gender, and age.

Model 3: Adjusted for gender, age, race, marital status, PIR, smoking, alcohol consumption, education level, PA, hypertension, CVD, diabetes, TC, HDL-C, AST, ALT, creatinine, total calcium, and uric acid.

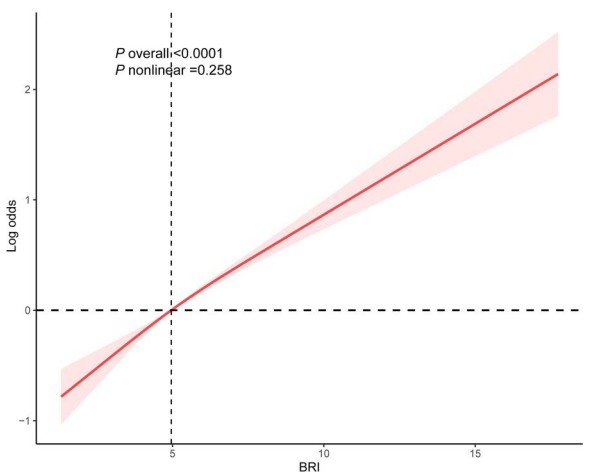

**Fig 2. Weighted RCS analysis of the association between the Body Roundness Index and osteoarthritis.** Red shaded areas represent the 95% Confidence Interval. Adjusted for gender, age, race, marital status, PIR, smoking, alcohol consumption, education level, PA, hypertension, CVD, diabetes, TC, HDL-C, AST, ALT, creatinine, total calcium, and uric acid. BRI: Body Roundness Index.

inflammatory response in KOA [44]. Additionally, adipose tissue secretes large amounts of adipokines, such as leptin, that promote cartilage degradation and activate the inflammatory response [45]. Moreover, obesity can disrupt the balance of the gut microbiota, leading to systemic inflammation that may impact distant organs, including the joints [46]. Obesity is also often associated with abnormalities in lipid metabolism, and fatty acid deposition in articular cartilage can further facilitate OA cartilage degradation [47].

Thus, accurately identifying and intervening in obese individuals is crucial for preventing and treating OA. Conventional body measurements cannot differentiate between fat and non-fat mass, failing to recognize the health damage caused by fat accumulation [19]. Based on WC and height, researchers constructed a new human morphological index, BRI, designed to provide a more comprehensive reflection of VAT and body fat volume [22]. The BRI has been demonstrated to be effective in evaluating CVD, nonalcoholic fatty liver disease, depression, and in predicting overall mortality in the general population [25,48,49]. Its diagnostic efficacy is superior to traditional body measurements for certain diseases, such as cholelithiasis, cancer, metabolic syndrome, and diabetes [20,50–52]. This study provides the first evidence of a positive

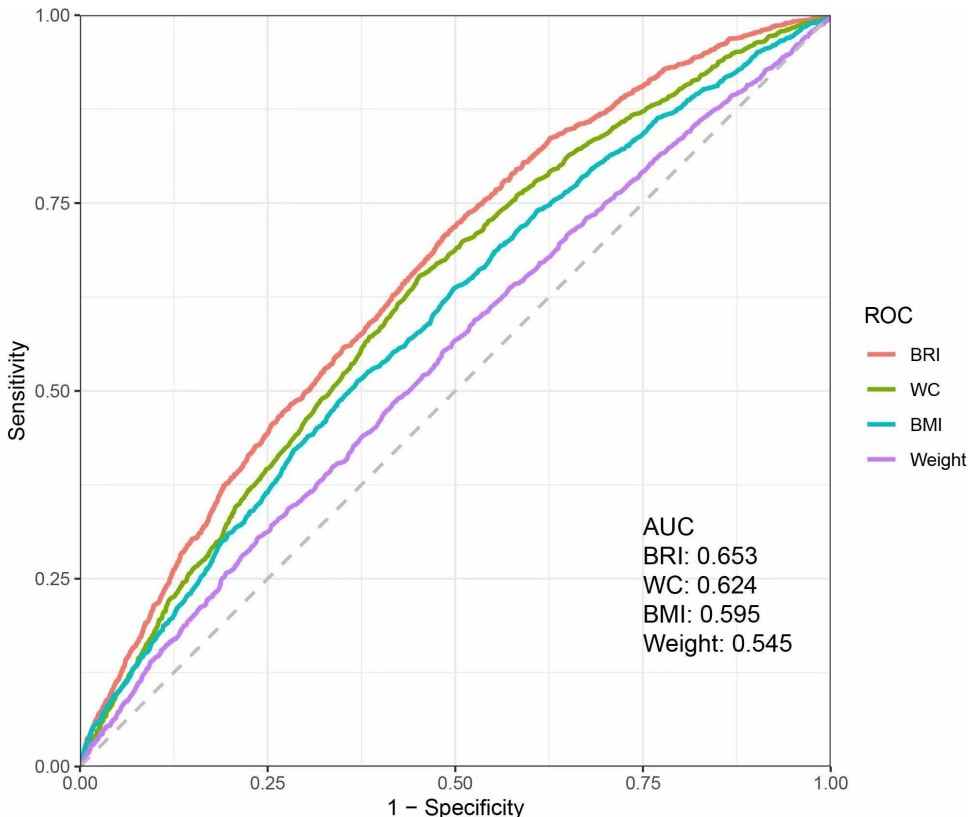

**Fig 3. Weighted ROC analysis comparing the diagnostic value of the Body Roundness Index and other anthropometric measures for osteoarthritis.**

correlation between BRI and the prevalence of OA. Although the diagnostic performance of BRI for OA, as indicated by the ROC analysis (AUC = 0.653), did not reach a high predictive threshold, it slightly outperformed traditional obesity indices such as WC, BMI, and weight. These findings suggest that BRI, potentially through its association with VAT, may serve as a complementary tool for assessing OA risk, particularly in populations where direct measurement of VAT is not feasible. The role of VAT in OA risk is further underscored by its unique metabolic and inflammatory properties. As an indicator of ectopic fat accumulation and metabolic disruption, VAT is a crucial site for metabolic processes [53]. Research indicates that visceral fat poses a greater health risk compared to subcutaneous fat [54]. This is because visceral adipocytes, as active endocrine organs, possess higher adipogenic and lipolytic activity than other adipose tissues and therefore produce more pro-inflammatory cytokines [27,28]. Numerous studies have established a strong link between VAT and OA. Specifically, Mendelian analysis has demonstrated that each unit increase in VAT elevates the risk of OA in the hip, knee, and spine by 40%, 79%, and 45%, respectively [55]. In KOA patients, the visceral fat thickness was markedly higher compared to that in healthy individuals, and this thickness related to the severity of OA [56]. Moreover, an accumulation of VAT has been linked to heightened levels of pain associated with OA [57]. Mechanistically, macrophage infiltration due to visceral fat deposition promotes the discharge of pro-inflammatory cytokines, further acting on the joints and leading to joint space narrowing and cartilage loss [58,59]. Visceral fat accumulation also inhibits lipocalin transcription and promotes leptin production, both of which exacerbate the progression of OA [60,61]. Given these findings, BRI should be actively monitored in individuals at high risk for OA, with particular attention to the adverse effects of VAT accumulation. Future

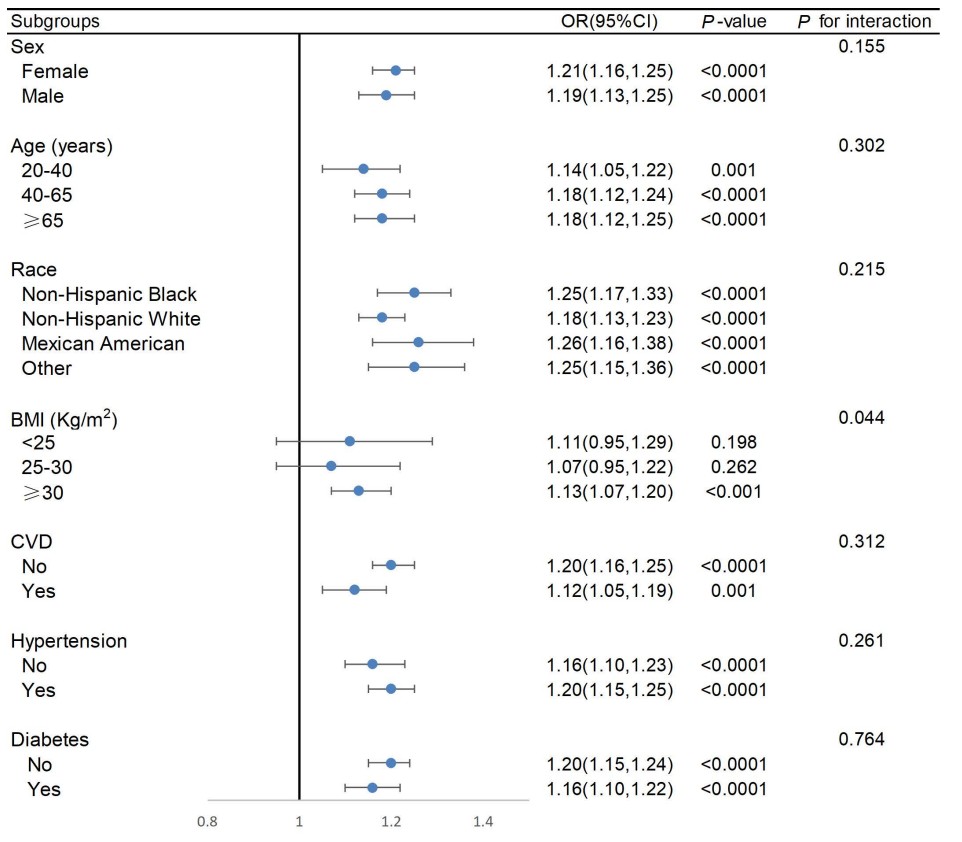

**Fig 4. Weighted subgroup analysis of the association between the Body Roundness Index and osteoarthritis.** Adjusted for gender, age, race, marital status, PIR, smoking, alcohol consumption, education level, PA, hypertension, CVD, diabetes, TC, HDL-C, AST, ALT, creatinine, total calcium, and uric acid, except the subgroup factors themselves. CVD: cardiovascular disease; BMI: body mass index; BRI: Body Roundness Index.

studies are needed to further elucidate the mechanisms linking BRI, VAT, and OA, as well as to validate the clinical utility of BRI in diverse populations.

Interestingly, our subgroup analyses showed a significant interaction between BMI and BRI, revealing that a potential positive association between BRI and OA was only established in the obese population with a BMI ≥ 30 kg/m². This may be explained by the fact that excessive weight gain exerts mechanical stress on the joints, which accelerates joint degeneration. However, when BMI is < 30 kg/m², the biomechanical changes due to weight are less significant [32]. Consistent with our view, Felson et al. also demonstrated that weight reduction was linked to a lower risk of OA in those with a BMI ≥ 25 kg/m², whereas this link was not evident in individuals with a BMI < 25 kg/m² [6]. Furthermore, the widespread inflammatory response in obese populations, coupled with excessive visceral fat accumulation, triggers cascading reactions that surpass the body's ability to compensate, leading to the development of OA [62,63].

This study has notable strengths, including its use of the NHANES database, which provides a substantial and high-quality sample. And all analyses were weighted, and the results can be generalized to the U.S. population. Additionally, we adjusted for potential covariates, in conjunction with previous studies, to mitigate the effects of confounding variables, enhancing the reliability and validity of the results. Nevertheless, this study has several limitations. Firstly, cross-sectional studies cannot establish causality, so future research should conduct relevant cohort studies to further validate the causality. Secondly, the results are based solely on noninstitutionalized U.S. residents, which limits their

generalizability to other populations. Since NHANES did not conduct specific examinations for CVD and OA, this study adopted diagnostic criteria referenced from other relevant studies, utilizing questionnaires administered by professional healthcare personnel to diagnose CVD and OA. This approach, while practical, may introduce recall bias. To address these limitations, it is essential to conduct large-scale prospective cohort studies in the future to further validate the potential relationship between BRI and OA.

## Conclusion

This study demonstrates that elevated BRI is associated with an elevated prevalence of OA, especially in obese populations with a BMI ≥ 30 kg/m². BRI is expected to be a simple and effective novel anthropometric indicator for predicting OA. We emphasize that for obese individuals with a BMI ≥ 30 kg/m², focusing on BRI and actively controlling visceral fat accumulation may offer significant advantages in lowering the risk of OA.

## Acknowledgments

We appreciate the National Health and Nutrition Examination Survey for supplying the data.

## Author contributions

**Conceptualization:** Yue Qiu, Huangyi Yin.

**Formal analysis:** Yang Cai.

**Funding acquisition:** Jun Yao.

**Investigation:** Yang Cai.

**Methodology:** Yue Qiu, Huangyi Yin.

**Software:** Yue Qiu, Huangyi Yin.

**Supervision:** Jun Yao.

**Validation:** Jinzhi Meng.

**Visualization:** Jinzhi Meng.

**Writing – original draft:** Yue Qiu, Huangyi Yin.

**Writing – review & editing:** Jun Yao.

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
