## [Decision Letter · Decision Letter 0]

PONE-D-24-44660
Association between the Body Roundness Index and osteoarthritis: a population-based study
PLOS ONE

Dear Dr. Yao,

Thank you for submitting your manuscript to PLOS ONE. After careful consideration, we feel that it has merit but does not fully meet PLOS ONE’s publication criteria as it currently stands. Therefore, we invite you to submit a revised version of the manuscript that addresses the points raised during the review process.

We look forward to receiving your revised manuscript.

Kind regards,

PLOS ONE

Additional Editor Comments:

Based on comments of reviewers, I recommond this manuscript minor revision.

Reviewers' comments:

Reviewer's Responses to Questions

**Comments to the Author**

1. Is the manuscript technically sound, and do the data support the conclusions?

Reviewer #1: Yes

Reviewer #2: Yes

2. Has the statistical analysis been performed appropriately and rigorously? 

Reviewer #1: Yes

Reviewer #2: Yes

3. Have the authors made all data underlying the findings in their manuscript fully available?

Reviewer #1: Yes

Reviewer #2: Yes

4. Is the manuscript presented in an intelligible fashion and written in standard English?

Reviewer #1: Yes

Reviewer #2: Yes

5. Review Comments to the Author

Reviewer #1: The authors performed population-based study to assess the Association between the Body Roundness Index and osteoarthritis. This study is interesting since it enlightens the general public about a new indicator of osteoarthritis. The study adapted a well-validated questionnaire and conducted the study in a planned manner with most parts well written. However, such as the use of ROC that yielded low diagnostic accuracies. I have a few comments to be addressed by the authors. If these comments are substantially addressed, I recommend publication of this manuscript.

Major Comments:

Abstract:

1. Line 11: “Obesity is closely linked to osteoarthritis (OA)” Kindly have opening sentence that show how Obesity is linked to OA.

2. Your ROC analysis had low performance and shouldn’t be emphasized here in abstract. You should ignore the fact that it can severe as a diagnostic indicator. ROC values less than 0.7 indicates that a variable is weak in prediction.

Introduction:

3. Kindly adopt the PLOS ONE referencing style.

Materials and Methods:

4. Authors should clearly state the study design.

Results:

1. Since Osteoarthritis is a main outcome of the study, kindly put the prevalence in a table.

2. Your ROC analysis had low performance and shouldn’t be emphasized here in the manuscript. Based on the AUC values close to 0.5/0.6, these parameters are guessers and cannot serve as good indicators of OA.

Discussions:

1. Well-written.

Reviewer #2: Very good paper with comprehensive information. However is it right to just ask the patient about arthritis, heart disease, or diabetes? I think that it may have more reliable way to confirm their OA or ...

6. PLOS authors have the option to publish the peer review history of their article (what does this mean?). If published, this will include your full peer review and any attached files.

Reviewer #1: **Yes: **Stephen Opoku

Reviewer #2: **Yes: **Laleh Abadi marand

---

## [Author Response · Author response to Decision Letter 1]

11 Feb 2025

Dear Editors and Reviewers:

Thank you for the editors' and reviewers' comments regarding our manuscript entitled

“Association between the Body Roundness Index and osteoarthritis: a population-based study” (Submission ID: PONE-D-24-44660). We greatly appreciate your positive feedback and valuable suggestions to enhance the quality of our manuscript. We have carefully reviewed the comments and made revisions, which we hope will meet your approval. The modifications are highlighted in different colors within the manuscript. Below are the main corrections and our responses to the editors' and reviewers' comments:

Editors' Comments:

1.Please ensure that your manuscript meets PLOS ONE's style requirements, including those for file naming. The PLOS ONE style templates can be found at https://journals.plos.org/plosone/s/file?id=wjVg/PLOSOne_formatting_sample_main_body.pdf and https://journals.plos.org/plosone/s/file?id=ba62/PLOSOne_formatting_sample_title_authors_affiliations.pdf

Response:

Thank you for the time and effort you have dedicated to reviewing our manuscript. We have carefully reviewed and modified the manuscript format according to the link you provided in order to meet the formatting requirements of PLOS ONE.

Response:

Thank you very much for your kind reminder. We have thoroughly reviewed the completeness and accuracy of the manuscript's references and can confirm that no retracted references have been cited.

Reviewer Comments:

Reviewer 1

Abstract:

1. Line 11: “Obesity is closely linked to osteoarthritis (OA)” Kindly have opening sentence that show how Obesity is linked to OA.

Response (this part of the revision is highlighted in Yellow):

We sincerely appreciate your valuable suggestion regarding the opening sentence. In response to your comment, we have revised the opening sentence to clearly illustrate the link between obesity and osteoarthritis. (line 12-13)

2. Your ROC analysis had low performance and shouldn’t be emphasized here in abstract. You should ignore the fact that it can severe as a diagnostic indicator. ROC values less than 0.7 indicates that a variable is weak in prediction.

Response:

We sincerely thank you for your careful review. In response to your comment regarding the low performance of the ROC analysis, we have removed the presentation of the ROC results from the abstract to avoid overemphasizing their limited diagnostic utility.

Introduction:

3. Kindly adopt the PLOS ONE referencing style.

Response:

Thank you for your valuable suggestion. In response to your comment, we have revised the reference format throughout the manuscript to fully comply with the PLOS ONE referencing style. We hope this revision meets the journal’s requirements and enhances the manuscript’s overall quality.

Materials and Methods:

4. Authors should clearly state the study design.

Response (this part of the revision is highlighted in Red):

Thank you for your thorough and thoughtful review. We appreciate your valuable suggestion regarding the study design. In response, we have revised the manuscript to include a detailed description of the study design, which can now be found on lines 101-114. We believe these changes have enhanced the clarity and transparency of our methodology.(line 101, 112-114)

Results:

1. Since Osteoarthritis is a main outcome of the study, kindly put the prevalence in a table.

Response (this part of the revision is highlighted in Blue):

Thank you for your valuable comment. As suggested, we have added the prevalence of osteoarthritis to Table 1. This inclusion provides a clearer and more comprehensive overview of the OA distribution within our study population. (Table 1)

2. Your ROC analysis had low performance and shouldn’t be emphasized here in the manuscript. Based on the AUC values close to 0.5/0.6, these parameters are guessers and cannot serve as good indicators of OA.

Response (this part of the revision is highlighted in Purple):

We sincerely appreciate your valuable feedback. We agree that the AUC values close to 0.5/0.6 indicate limited predictive performance. In accordance with your suggestion, we have revised the manuscript to de-emphasize the ROC results. Specifically, we have rephrased the description of the ROC analysis in the Results section to focus on the objective presentation of the data without overstating its diagnostic significance. Additionally, in the Discussion section, we have revised the interpretation of these results to provide a more balanced perspective and emphasized the need for further research to validate these findings. Thank you again for your insightful comment, which has helped us improve the clarity and rigor of our manuscript. (line 199-203, 220-221, 278-287, 301-305)

Discussions:

1. Well-written.

Response:

We sincerely appreciate your constructive feedback.

Reviewer 2

1.Very good paper with comprehensive information. However is it right to just ask the patient about arthritis, heart disease, or diabetes? I think that it may have more reliable way to confirm their OA or ...

Response (this part of the revision is highlighted in Green):

Thank you very much for your thorough review and valuable suggestions. NHANES is a cross-sectional study, and the detailed examination of heart disease and osteoarthritis is still under development. In previous NHANES studies, osteoarthritis was diagnosed based on results from questionnaires administered by medical professionals [1-5], and a similar diagnostic approach was used for cardiovascular diseases [6-9]. This suggests that self-reported osteoarthritis and cardiovascular diseases hold a certain degree of accuracy and research value. To improve the rigor of the manuscript, we have addressed potential biases arising from this diagnostic method in the discussion section, highlighting the need for large-scale prospective cohort studies to further validate the relationship between BRI and osteoarthritis.

Moreover, after carefully considering your invaluable feedback and reviewing previous NHANES studies, we have revised the diagnostic criteria for diabetes in the manuscript [10-12]. Specifically, a participant will be considered to meet the diagnostic criteria for diabetes if they satisfy any of the following conditions: glycated hemoglobin (HbA1c) ≥ 6.5%, fasting blood glucose ≥ 7.0 mmol/L, 2-hour postprandial blood glucose ≥ 11.1 mmol/L, currently using antidiabetic medications or insulin, or self-reporting a history of diabetes. Based on this revision, we have updated both the "Methods" and "Results" sections of the manuscript. We would like to emphasize that these changes do not alter the overall conclusions of the study.

Once again, thank you for your insightful feedback. (line 23-26, 139-142, 183-189, 211-213, 325-331, Table 1, Table 2, Figure 2, Figure 4)

References：

[1] Chen L, Zhao Y, Liu F, et al. Biological aging mediates the associations between urinary metals and osteoarthritis among U.S. adults. BMC Med. 2022;20(1):207. Published 2022 Jun 17. doi:10.1186/s12916-022-02403-3

[2] Zhao D, Shen S, Guo Y, et al. Flavan-3-ol monomers intake is associated with osteoarthritis risk in Americans over 40 years of age: results from the National Health and Nutritional Examination Survey database. Food Funct. 2024;15(13):6966-6974. Published 2024 Jul 1. doi:10.1039/d3fo04687g

[3] Huang J, Han J, Rozi R, et al. Association between lipid accumulation products and osteoarthritis among adults in the United States: A cross-sectional study, NHANES 2017-2020. Prev Med. 2024;180:107861. doi:10.1016/j.ypmed.2024.107861

[4] Liu Y, Song F, Liu M, et al. Association between omega-3 polyunsaturated fatty acids and osteoarthritis: results from the NHANES 2003-2016 and Mendelian randomization study. Lipids Health Dis. 2024;23(1):147. Published 2024 May 17. doi:10.1186/s12944-024-02139-4

[5] Xue H, Zhang L, Xu J, et al. Association of the visceral fat metabolic score with osteoarthritis risk: a cross-sectional study from NHANES 2009-2018. BMC Public Health. 2024;24(1):2269. Published 2024 Aug 21. doi:10.1186/s12889-024-19722-0

[6] Dang K, Wang X, Hu J, et al. The association between triglyceride-glucose index and its combination with obesity indicators and cardiovascular disease: NHANES 2003-2018. Cardiovasc Diabetol. 2024;23(1):8. Published 2024 Jan 6. doi:10.1186/s12933-023-02115-9

[7] Zhang Q, Xiao S, Jiao X, Shen Y. The triglyceride-glucose index is a predictor for cardiovascular and all-cause mortality in CVD patients with diabetes or pre-diabetes: evidence from NHANES 2001-2018. Cardiovasc Diabetol. 2023;22(1):279. Published 2023 Oct 17. doi:10.1186/s12933-023-02030-z

[8] Han L, Wang Q. Association of Dietary Live Microbe Intake with Cardiovascular Disease in US Adults: A Cross-Sectional Study of NHANES 2007-2018. Nutrients. 2022;14(22):4908. Published 2022 Nov 20. doi:10.3390/nu14224908

[9] Zhang H, Tian W, Qi G, Zhou B, Sun Y. Interactive association of the dietary oxidative balance score and cardiovascular disease with mortality in older adults: evidence from NHANES. Food Funct. 2024;15(11):6164-6173. Published 2024 Jun 4. doi:10.1039/d4fo01515k

[10] Liao J, Wang L, Duan L, et al. Association between estimated glucose disposal rate and cardiovascular diseases in patients with diabetes or prediabetes: a cross-sectional study. Cardiovasc Diabetol. 2025;24(1):13. Published 2025 Jan 13. doi:10.1186/s12933-024-02570-y

[11] Zhong J, Zhang Y, Zhu K, et al. Associations of social determinants of health with life expectancy and future health risks among individuals with type 2 diabetes: two nationwide cohort studies in the UK and USA. Lancet Healthy Longev. 2024;5(8):e542-e551. doi:10.1016/S2666-7568(24)00116-8

[12] Liu C, Liang D. The association between the triglyceride-glucose index and the risk of cardiovascular disease in US population aged ≤ 65 years with prediabetes or diabetes: a population-based study. Cardiovasc Diabetol. 2024;23(1):168. Published 2024 May 13. doi:10.1186/s12933-024-02261-8

We tried our best to improve the manuscript and made some changes in the manuscript.

These changes will not influence the content, conclusion and framework of the paper. We appreciate for editors/reviewers’ warm work earnestly, and hope that the correction will meet with approval.

Once again, thank you very much for your comments and suggestions.

Yours sincerely,

Yue Qiu

Corresponding author:

Name: Jun Yao

E-mail: yaojun800524@126.com

---

## [Decision Letter · Decision Letter 1]

Association between the Body Roundness Index and osteoarthritis: a population-based study

PONE-D-24-44660R1

Dear Dr. Yao,

We’re pleased to inform you that your manuscript has been judged scientifically suitable for publication and will be formally accepted for publication once it meets all outstanding technical requirements.

Kind regards,

Li-Da Wu

Academic Editor

PLOS ONE

Additional Editor Comments (optional):

Reviewers' comments:

Reviewer's Responses to Questions

**Comments to the Author**

1. If the authors have adequately addressed your comments raised in a previous round of review and you feel that this manuscript is now acceptable for publication, you may indicate that here to bypass the “Comments to the Author” section, enter your conflict of interest statement in the “Confidential to Editor” section, and submit your "Accept" recommendation.

Reviewer #1: All comments have been addressed

Reviewer #2: All comments have been addressed

2. Is the manuscript technically sound, and do the data support the conclusions?

Reviewer #1: Yes

Reviewer #2: Yes

3. Has the statistical analysis been performed appropriately and rigorously? 

Reviewer #1: Yes

Reviewer #2: Yes

4. Have the authors made all data underlying the findings in their manuscript fully available?

Reviewer #1: Yes

Reviewer #2: Yes

5. Is the manuscript presented in an intelligible fashion and written in standard English?

Reviewer #1: Yes

Reviewer #2: Yes

6. Review Comments to the Author

Reviewer #1: All my comments were appropriately addressed, however, I mistakenly said a table instead of a figure in the below comment, which I believe the authors should readdress

Results:1. Since Osteoarthritis is a main outcome of the study, kindly put the prevalence in a table.

Although the authors added this to table 1 based on my comment (table), it should be a figure. Just have a figure (bar graph) showing the percentage of OA and non-OA, so it is easy for readers to capture your main outcome.

since OA-yes is the same as OA and OA-no is the same as non-OA. I am happy with the remaining repose and do not need to reassess this manuscript again.

Reviewer #2: Thank you for your complete responses to all items. all comments have been addressed and if the other reviewers have accepted the responses, it is ready to publish.

7. PLOS authors have the option to publish the peer review history of their article (what does this mean?). If published, this will include your full peer review and any attached files.

Reviewer #1: **Yes: **Stephen Opoku

Reviewer #2: No

---

## [Editor Report · Acceptance letter]

PONE-D-24-44660R1

PLOS ONE

Dear Dr. Yao,

I'm pleased to inform you that your manuscript has been deemed suitable for publication in PLOS ONE. Congratulations! Your manuscript is now being handed over to our production team.

Kind regards,

on behalf of

Dr. Li-Da Wu

Academic Editor

PLOS ONE